# A Combined Risk Score Model to Assess Prognostic Value in Patients with Soft Tissue Sarcomas

**DOI:** 10.3390/cells11244077

**Published:** 2022-12-16

**Authors:** Zihua Li, Zhengwei Duan, Keyao Jia, Yiwen Yao, Kaiyuan Liu, Yue Qiao, Qiuming Gao, Yunfeng Yang, Guodong Li, Anquan Shang

**Affiliations:** 1Department of Laboratory Medicine, Shanghai Tongji Hospital, School of Medicine, Tongji University, Shanghai 200065, China; 2Department of Orthopedics, Shanghai Tenth People’s Hospital, School of Medicine, Tongji University, Shanghai 200072, China; 3Department of Orthopedics, Shanghai Tongji Hospital, School of Medicine, Tongji University, Shanghai 200065, China; 4The First School of Clinical Medicine, Southern Medical University, Guangzhou 510515, China; 5Department of Internal Medicine V-Pulmonology, Allergology, and Respiratory Intensive Care Medicine, Saarland University Hospital, 66424 Homburg, Germany; 6Department of Laboratory Medicine, The Second People’s Hospital of Lianyungang & The Oncology Hospital of Lianyungang, Lianyungang 222006, China

**Keywords:** cuproptosis, prognosis, immune infiltration, soft tissue, biomarker

## Abstract

A study by Tsvetkov et al. recently published a proposed novel form of copper-induced cell death in *Science*; however, few studies have looked into the possible mechanism in soft tissue sarcoma (STS). Herein, this study sought to investigate the function of cuproptosis-related genes (CRGs) in the development of tumor-associated immune cells and the prognosis of sarcoma. Herein, this study aimed to explore the role of cuproptosis-related genes (CRGs) in the development, tumor-associated immune cells, and the prognosis of sarcoma. Methods: The prognostic model was established via the least absolute shrinkage and selection operator (LASSO) algorithm as well as multivariate Cox regression analysis. The stromal scores, immune scores, ESTIMA scores, and tumor purity of sarcoma patients were evaluated by the ESTIMATE algorithm. Functional analyses were performed to investigate the underlying mechanisms of immune cell infiltration and the prognosis of CRGs in sarcoma. Results: Two molecular subgroups with different CRG expression patterns were recognized, which showed that patients with a higher immune score and more active immune status were prone to have better prognostic survival. Moreover, GO and KEGG analyses showed that these differentially expressed CRGs were mainly enriched in metabolic/ions-related signaling pathways, indicating that CRGs may have impacts on the immune cell infiltration and prognosis of sarcoma via regulating the bioprocess of mitochondria and consequently affecting the immune microenvironment. The expression levels of CRGs were closely correlated to the immunity condition and prognostic survival of sarcoma patients. Conclusions: The interaction between cuproptosis and immunity in sarcoma may provide a novel insight into the study of molecular mechanisms and candidate biomarkers for the prognosis, resulting in effective treatments for sarcoma patients.

## 1. Introduction

Soft tissue sarcoma (STS), which primarily develops from malignant mesenchymal cells and has about 175 distinct subtypes, accounts for 1% of all adult cancers. It is highly heterogeneous in terms of anatomical location, molecular features, and distinctive prognosis [1,2]. The easy occurrence and high rate of metastasis are additional features of STS. Despite the current state of advanced therapy, such as surgical tumorectomy, chemotherapy, pre and postoperative neoadjuvant chemotherapy, radiotherapy, and immunotherapy, the therapeutic outcomes of STS treatments are still far from satisfactory [3,4]. In order to enhance both the quality of life for STS patients and the overall survival rate, great importance should be attached to early detection, diagnosis, and intervention to reduce recurrence and mortality [5]. Over the decades, research concerning tumor molecular characteristics has demonstrated cellular and molecular mechanisms that promote tumorigenesis and progression. Therefore, it is essential to develop cutting-edge therapy strategies and to comprehend the mechanisms behind the development and progression of STS.

In the quest to understand the mechanisms behind cancer and development, microelement metabolism research has recently drawn a lot of interest [6]. Iron is not only a fundamental microelement for life, but it is also essential for multiple biological processes, such as DNA synthesis, energy transfer, erythropoiesis, and cell signal pathways [7]. Cu is essential in eukaryotes for oxygen metabolism, oxygen radical detoxification, and Fe uptake [8]. Additionally, research has demonstrated that copper (Cu) functions as a dynamic signaling component with a considerable influence on a variety of biological processes, namely lipolysis, cellular proliferation, autophagy, and brain activity [9]. As a result, given that the level of Cu in serum and malignant tissues is elevated in comparison to their healthy counterparts, emphasis should be placed on the potential applications of Cu in cancer cells, including cell proliferation, angiogenesis, and metastasis. Disturbance of Cu homeostasis can induce cell death through multiple approaches, including Cu-induced apoptosis, autography, and paraptosis, among others. Consequently, regulating intracellular copper ion concentration could serve as a novel target for the therapy of tumors.

According to research reported in *Science*, intracellular Cu accumulation activates the aggregation of mitochondrial lipoylated proteins and the destabilization of Fe–S cluster proteins, causing a novel type of cell death termed cuproptosis [10]. Unlike ferroptosis, Cu-induced cell death relies on the Fenton reaction to generate ROS and upregulate the expression of genes related to apoptosis or to result in mitochondrial malfunction, which activates the apoptosis pathway. Given the tight correlation between copper homeostasis and tumorigenesis, it is vital to recognize CRGs, elucidate their potential mechanism in the progression of STS, and investigate the correlation between CRGs and the immune microenvironment. Nevertheless, few studies have examined the function of CRGs in STS etiology and prognosis as well as their clinical signature and immune microenvironment [11,12]. In order to ascertain whether CRGs affect STS prognosis and immune infiltration while also attempting to investigate its potential mechanism or other mechanisms through which cuproptosis impacts STS, our study combines a number of bioinformatics techniques with qRT-PCR validation of tissues from STS patients divided into high-risk and low-risk groups. The CRGs identified in our study were also used to build a prognostic model, indicating the immune microenvironment, metabolic rewiring, and the mutation landscape, emphasizing the prominent role of CRGs in tumorigenesis and revealing that their expression may regulate tumor-associated immune cell infiltration in STS.

## 2. Materials and Methods

### 2.1. Collection of Sarcoma Datasets

The Caner Genome Atlas (TCGA) data were obtained from the Genomic Data Commons (GDC) Data Portal including RNA-sequencing data (FPKM values) of STS in TCGA-SARC cohort (https://xenabrowser.net/datapages/?cohort=GDC%20TCGA%20Sarcoma%20 (SARC) & removeHub=https%3A%2F%2Fxena.treehouse.gi.ucsc.edu%3A443) accessed on 5 May 2022.

### 2.2. Identification of Prognostic CRGs

We used the Gene Set Variation Analysis (GSVA, version 1.22.4) algorithm to explore the activity variation of biological process (BP) terms in GO analysis and create the cuproptotic risk score based on CRGs defined by the research published in Science by Tsvetkov et al., which includes FDX1, LIPT1, LIAS, DLD, DBT, GCSH, DLST, DLAT, PDHA1, PDHB, SLC31A1, ATP. A weighted gene co-expression network analysis (WGCNA, version 1.70) was conducted on TCGA-SARC according to the cuproptotic risk score (Appendix A) to investigate the hub genes. Topologically resemble modules were combined into a neo-cluster, and correlation analysis was used to examine the correlation of module genes and phenotypes. Modules with comparatively strong positive correlations and a cuproptotic risk score were designated, and then the top 200 genes in hubturquois modules were chosen using cytoHubba in cytoscape. The survival connection with CRGs was evaluated sequentially using least absolute shrinkage and selection operator (LASSO) regression and ridge regression, and CRGs with comparatively larger weight were identified to construct the prognosis of STS patients [13,14,15]. LASSO Cox regression was applied to explore the optimal CRGs, and the formula for risk scores is presented as follows, according to the result:(1)$${\mathrm{Risk}\;\mathrm{Score}}_{j\;}=\overset{n}{\underset{i=1}{\sum }}\propto_{i}k_{ij}$$

### 2.3. Functional Analyses and Mechanism Investigation

To further investigate the potential impact of CRGs on promoting tumorigenesis in STS, functional enrichment analyses were conducted. GO and KEGG analyses were conducted to recognize the biological functions of these differentially expressed CRGs. Three terms made up the GO categories: cellular component, biological process, and molecular function (MF) (CC). The ‘clusterProfiler’ (version 4.4.4) and ‘limma’ (version 3.52.3) packages in the R software were employed to conduct all enrichment analyses based on background genes from the GO or KEGG databases. [16,17].

### 2.4. Immune Infiltration Analysis

CIBERSORT was applied to investigate the levels of tumor immune-infiltration, and the tumor immune estimation resource (TIMER) (http://timer.cistrome.org) accessed on 5 May 2022 web server was used to investigate TME [18,19]. The tumor immune estimation resource (TIMER), which consists of molecular characterization of 32 types of cancer and six types of immune cells, was applied to investigate the correlation between five hub CRGs and immune cell types. To measure the relationship between tumor purity and riskscores, the expression data (ESTIMATE) algorithm, which includes an immune score, stromal score, and ESTIMATE score, was used to estimate the number of stromal and immune cells in malignant tumors [20].

### 2.5. Exploration the Relationship between TME and Risk Scores

The relationship between risk scores and immune responses was investigated by the TIMER database and enumerated by several methods [21,22]. The ssGSEA, based on the expression level of 28 immunity-associated signatures (http://cis.hku.hk/TISIDB/download.php) accessed on 5 May 2022 algorithm was then conducted to evaluate the differences in immune function and immune infiltration in the high and low risk groups. Additionally, the CIBERSORT algorithm (1000 permutations) was applied to evaluate the Pearson correlation coefficients between risk scores and tumor immune infiltration and the percentage of infiltration in each sample, which showed statistical significance. In the high- and low-risk groups, we compared the expression discrepancy of representative immune checkpoint molecules by using a Wilcoxon rank-sum test [23]. To quantify the correlation of tumor purity and risk scores, the estimation of stromal and immune cells in malignant tumors that uses expression data (ESTIMATE) algorithm containing an immune score, stromal score, and ESTIMATE score was applied. A univariate COX regression analysis was conducted to recognize prognosis-related CRGs in STS patients’ data obtained from the TCGA database.

### 2.6. Mutation Landscape Analysis

RNA-sequencing expression profiles and corresponding clinical information for SARC were downloaded from the TCGA dataset (https://portal.gdc.com) accessed on 5 May 2022. The landscape of top mutated genes between the high- and low-risk groups was presented with mutation types and frequencies, and data analysis was performed by the maftools package (Version 2.12.0).

### 2.7. Clinical Specimens

Data of 33 STS patients who underwent radical resection at the Shanghai Tongji Hospital and Shanghai Tenth People’s Hospital of Tongji University. None of the patients underwent preoperative radiotherapy or chemotherapy. According to TNM classification, we divided STS patients enrolled into a high-risk group and a low-risk group [24], including 20 high-risk and 13 low-risk patients. The clinicopathological details are presented in Additional File: Appendix A. Ethical approval for the study was granted by the Clinical Research Ethics Committee in our hospital, and all participants included in our study provided written informed consents.

### 2.8. Quantitative Reverse Transcription Polymerase Chain Reaction (qRT-PCR)

Total RNA was extracted from tissues of STS patients using the Trizol reagent (Invitrogen, Waltham, MA, USA), following the manufacturer’s protocol. Then, the expression levels of mRNA from tissues were determined by qRT-PCR (ABI 7500 real-time PCR system, Applied Biosystems, California, CA, USA) using SYBR Green Mix (Bio-Rad, Hercules, California, USA). The relative expression levels of mRNAs were enumerated using the 2^−ΔΔCt^ method, and GAPDH (Glyceraldehyde-3-Phosphate Dehydrogenase) was set as the internal reference for mRNA. The primer sequence is presented in Appendix A.

### 2.9. Statistical Analyses

Univariate Cox regression analysis and survival analysis were conducs’sted using R software (version 3.6.3) to evaluate prognostic and survival-related genes for the Kaplan–Meier curve. Additionally, multivariate COX regression analysis was conducted to identify and establish the prognostic model using the prognostic prediction gene. As for data complied with the normal distribution, a Student’s *t* test and one-way analysis of variance (ANOVA) were conducted, and a *p* value less than 0.05 can be regarded as a significantly statistical difference [25].

## 3. Result

### 3.1. Identification of the Cuproptosis Subtypes and Prognostic CRGs

In our analysis, a total of 255 samples of STS patients with complete survival information were reserved (Appendix A). The soft threshold in WGCNA of 5 was determined by calculating the scale-free model fit and mean connectivity. Various module genes were re-clustered by the utilization of a topological similarity strategy, as a result of which genes were fitting into fewer modules, as presented in Figure 1A,B. Each gene module was assigned a color value once the gene clustering results were cut to obtain different gene modules. The results were then added to the clustering tree, and distinction was achieved by using the previously allocated color (Figure 1C). The relationships between modules and cuproptotic risk score indicated that the modules in black (r = −0.14, *p* = 0.02), brown (r = −0.25, *p* < 0.001), green (r = 0.28, *p* < 0.001), magenta (r = 0.23, *p* < 0.001), pink (r = −0.18, *p* = 0.004), red (r = 0.15, *p* = 0.01), royal blue (r = 0.14, *p* = 0.02), saddle brown (r = 0.18, *p* = 0.004), salmon (r = 0.13, *p* = 0.03), steel blue (r = 0.16, *p* = 0.01), tan (r = 0.13, *p* = 0.03), turquoise (r = 0.36, *p* < 0.001) had comparatively strong correlations with cuproptosis (Figure 1D). The red-color module demonstrated a marginally positive relationship with survival status, and the related genes were applied to obtain further analysis in the next steps. BP participated in peptidyl-lysine modification, chromatin organization, histone modification, and protein acylation. MF mainly regulated the production of transcription regulator complexes, nuclear specks, and proteins, as well as histone acetyltransferases. CC were primarily upregulated in transcription coregulator activity, RNA polymerase II-specific DNA-binding transcription factor binding, and histone binding. KEGG-based analysis indicated the overexpressed genes were mainly involved in lysine degradation, carbon metabolism, thyroid hormone signaling pathway, cell cycle, glucagon signaling pathway, citrate cycle (TCA cycle), and pyruvate metabolism (Figure 1E,F).

### 3.2. Determination of CRGs and Validation

The significance of prognostic genes in the model was determined using Cox regression analysis, and the result indicated that there are significant differences in LIPT1(hazard ratio, HR, 0.610; 95% confidence interval, CI, 0.404–0.920; *p* = 0.018), GCSH (HR:1.76, 95% CI: 1.160–2.692, *p* = 0.008), ATP7B (HR: 1.766, 95% CI: 1.174–2.655, *p* = 0.006), APAF1 (HR: 1.428, 95% CI: 1.034–1.974, *p* = 0.031), NUP153 (HR: 1.465, 95% CI: 1.088–1.974, *p* = 0.012), NCOA6 (HR: 1.754 95% CI: 1.115–2.760, *p* = 0.015), PRPF4B (HR: 1.725, 95% CI: 1.164–2.553, *p* = 0.007) and TAF2 (HR: 1.355, 95% CI: 1.004–1.828, *p* = 0.047), Figure 2A. LASSO regression was applied to sort the weights in the prognosis according to 5 survival-related CRGs based on the optimal value of lambda (Figure 2B,C). Therefore, the risk scores could be calculated based on the formula: risk score = −0.4795 × Exp (LIPT1) + 0.3682 × Exp (GCSH) + 0.1190 × Exp (ATP7B) + 0.1771 × Exp (NCOA6) + 0.33591× Exp (PRPF4B), where Exp (∙) indicated the expression of CRGs. Patients were divided into the high-risk and low-risk groups depending on the median value of their risk scores. A KM analysis was conducted to show that STS patients experienced a significant discrepancy in survival outcomes in the high-risk group and in the low-risk group (*p* = 0.0048). Additionally, risk curves revealed that patients in the low-risk group presented better survival outcomes (Figure 3A–D).

### 3.3. GO Terms and KEGG Pathway Analysis of the Differentially Expressed Genes in High- and Low-Risk Group

The volcano plot showed the DEGs in the high- and low-risk groups of patients with sarcoma according to screening criteria (false discovery rate (FDR) < 0.05, |log fold change| > 0.5). A total of 18,731 genes were attained in our research, including 1039 upregulated and 737 downregulated. (Figure 4A) The detailed information on genes was summarized and presented in Appendix A. Then, the possible biological functions and processes of DEGs were demonstrated using GO and KEGG functional analyses. The external subject’s dataset, which was separated into two sub-sets, was revealed by the PLS-DA model and constructed on values (Figure 4B). The related genes were applied for further analysis in the following steps. Through plasma-membrane adhesion molecules, BP played a role in the development of muscle tissue, striated muscle tissue, muscle organs, cell fate commitment, and cell-cell adhesion. MF mainly regulated the synaptic and postsynaptic membranes, including intrinsic and integral. CC were primarily upregulated in gated channel activity, passive transmembrane transporter activity, channel activity, ion channel activity, and cation channel activity. KEGG-based analysis suggested that the overexpressed genes were mainly involved in neuroactive ligand-receptor interaction, cytokine-cytokine receptor interaction, calcium signaling pathway, hematopoietic cell lineage, and cell adhesion molecules. (Figure 4C,D).

### 3.4. Biological Characteristics and TME Investigation

We investigated the two distinctive patterns of immune infiltration in the high-risk and low-risk groups. (Figure 5A,B) Additionally, the associations between risk and immune infiltration were estimated by the CIBERSORT algorithm. The proportions of immune infiltration from 22 cell types in the two risk subtypes were presented in a bar plot (Figure 5C). According to the consequences, plasma cells, T cell gamma delta, macrophages, mast cells resting, and T cells CD8, 4 memory resting, etc., were notable, which reveal lower infiltration enrichment in the high-risk subtype while the mast cells resting presented a higher level in the low-risk subtype and macrophages M0 presented higher levels in the high-risk subtype (Figure 5D). Figure 6 shows the specific correlation between different types of immune cells and risk scores. In order to investigate the populations of immune cells between the signature gene levels, the levels of tumor immune infiltration were conducted, which are presented in Figure 7.

### 3.5. Hub CRGs Immune Infiltration Analysis

According to a report, immune cell infiltration and the expression of the five hub CRGs were correlated, and this finding had an effect on the prognosis for STS. The results showed that LIPT1 was negatively corelated with macrophages (partial. cor = −0.16, *p* = 1.38 × 10^−2^) and dendritic cell (partial. cor = −0.188, *p* = 3.40 × 10^−3^). Similarly, GCSH was negatively corelated with dendritic cell (partial. cor = −0.179, *p* = 5.27 × 10^−3^) and ATP7B was negatively corelated with CD4^+^ T cells (partial. cor = −0.198, *p* = 2.10 × 10^−3^), macrophages (partial. cor = −0.137, *p* = 3.55 × 10^−2^), and dendritic cells (partial. cor = −0.283, *p* = 8.29 × 10^−6^). NCOA6 was negatively corelated with CD4^+^ T cells (partial. cor = −0.285, *p* = 7.68 × 10^−6^), macrophages (partial. cor = −0.133, *p* = 4.21 × 10^−2^), and dendritic cells (partial. cor = −0.24, *p* = 1.66 × 10^−4^). PRPF4B was negatively corelated with CD4^+^ T cells (partial. cor = -−0.309, *p* = 1.12 × 10^−6^), macrophages (partial. cor = −0.177, *p* = 6.49 × 10_−3_), and dendritic cells (partial. cor = −0.318, *p* = 4.70 × 10^−7^). Additionally, the results indicated that tumor purity was only significantly correlated with LIPT1 (cor = 0.176, *p* = 5.84 × 10^−3^), NCOA6 (cor = 0.302, *p* = 1.43 × 10^−6^) and PRPF4B (cor = 0.188, *p* = 3.06 × 10^−3^), respectively. (Figure 7A–E).

### 3.6. Correlation between Molecular Features and Risk

Sequentially, ESTIMATE scores were conducted to evaluate the tumor purity, including immune scores and stromal scores. The results showed that the risk scores were negatively correlated with stromal scores (r = −0.27, *p* < 0.001), immune scores (r = −0.37, *p* < 0.001), and ESTIMATE scores (r = −0.36, *p* < 0.001), Figure 8A–F. A Wilcoxon rank-sum test was performed to compare a series of immune checkpoint expression level in the high-risk and low-risk groups, which indicated that there was significant difference in PDCD1, CD274, CTLA4, CD86, PDCD1LG2, LAG3, HAVCR2, and CXCL9 (*p* < 0.05), Figure 6G. Sequentially, the correlation of cytolytic activity (CYT, which represents tumor-infiltrating T lymphocyte activity and is related to prognosis) score and risk score was investigated, and the results showed that risk scores had a significantly negative link with CYT score (r = −0.31, *p* < 0.001, Figure 9A). Additionally, there was a significant difference in tumor immune dysfunction and exclusion (TIDE) between the high- and low-risk groups and significant difference in risk score comparing the TRUE and FALSE group (*p* < 0.05, Figure 9B–D), where the responder of FALSE and TRUE stand for the sensitivity to immune checkpoint therapy.

### 3.7. Mutation Landscape Analysis

Additionally, mutation profiles of risk were conducted, and the results indicated that TP53, ATRX, and MUC16 ranked the highest in mutation proportion with the most missense mutation, followed by TTN in the high-risk group, while TP53, TTN, RB1, and ATRX ranked the highest in mutation proportion with the most missense mutation in the low-risk group. Patients in the low-risk group ranked higher in frequency of TP53 mutations (Figure 10A–N).

### 3.8. qRT-PCR Validations in Sarcoma Patients’ Tissues

We first investigated the expression level of CRGs between the high- and low-risk groups by bioinformatic analysis. The results showed that there were significant differences in LIPT1, DLD, DBT, GCSH, DLAT, PDHA1, SCL31A1, ATP7A, and ATP7B between the high- and low-risk groups (*p* < 0.05, Figure 11A). To further confirm the results, we selected qRT-PCR-verified CRGs to validate their expression patterns in high-risk and low-risk patients’ tissues (patients’ characteristics and details were presented in the Method section and Appendix A. Consequently, based on the result of the LASSO regression, we performed qRT-PCR to validate the expression level of DEGs, and the results demonstrated that there was a significant difference in the expression of LIPT1, GSCH, ATP7B, NCOA6, and PRPF4B, consistent with the bioinformatic analysis results (Figure 11B). Additionally, nine out of nine CRGs, including LIPT1, DLD, DBT, GCSH, DLAT, PDHA1, SLC31A1, ATP7A, and ATP7B, were shown to have concurrent expression trends in accordance with the above results. (Figure 11C)

## 4. Discussion

Due to the rare incidence rate and diverse origins of STS, precise diagnosis and prediction of its prognosis have become strikingly difficult [26,27,28]. Several therapeutic methods have been applied for STS patients, including surgical, chemotherapy, radiotherapy, and immunotherapy, but the overall survival of STS patients could not reach a satisfactory level [29,30,31,32]. At present, bioinformatic analysis, such as weighted gene co-expression network analysis (WGCNA), has been applied for investigating novel diagnostic and prognostic genes for unveiling the underlying mechanisms of cancer. Moreover, in order to determine the mechanism of inhibition or promotion of tumors, LASSO Cox regression was conducted to enumerate the coefficients of each gene and establish a prognostic risk system, which could serve as an independent prognostic factor. In our study, we aimed to establish a risk model with CRGs and predict prognosis in STS patients, as well as investigate the mutation landscape and tumor-associated immune cell infiltration.

Multiple precisely regulated programmed cell deaths, including apoptosis, pyroptosis, and necroptosis related to mitochondrial shrinkage and the accumulation of ROS, have been reported in published studies. Microelements could serve a vital role in cell development, and they can be regarded as indispensable for tumorigenesis. Nowadays, iron triggers cell death, which has become prevalent, and copper can serve as a necessary micronutrient for plenty of indispensable biological processes [33,34]. Nevertheless, unbalanced copper homeostasis may trigger cell death, and evidence referring to a potential mechanism has been reported. An article published in *Science* by Tsvetkov et al. [13] reported a novel type of cell death, copper-dependent cell death, as cuproptosis, indicating that cell death was due to copper accumulation resulting in toxicity tightly correlated with mitochondrial activity. Therefore, it is vital to make a further investigation into the correlation of molecular pathomechanism of cuproptosis-related genes with STS tumorigenesis.

Additionally, the results of further investigation demonstrated that the two subtypes also presented heterogeneity in prognosis, mutation patterns, immune cell infiltration, and response to immune checkpoint therapy. Sequentially, the scoring system established in our study, also named the cuproptosis score, was also conducted to estimate the CRGs’ expression levels in each STS patient, which was effective in distinguishing between high- and low-risk STS patients. In this study, a total of five CRGs, including LIPT1, GCSH, ATP7B, NCOA6, and PRPF4B, were chosen for establishing the prognostic risk model, which also verified the expression level by qRT-PCR in STS patients’ tissue. To our knowledge, LIPT1, NCOA6, and PRPF4B have not been elucidated in sarcoma. A study published by Gu established a risk score based on the expression of five genes, including GSCH to predict survival in STS, which effectively distinguishes between high- and low-risk STS patients [34], and GSCH plays an important role in breast and papillary thyroid cancer [35]. Additionally, Kakuda reported that ATP7B suppression can promote the platinum resistance of uterine leiomyosarcoma cells in vitro and in vivo with CuSO_4_ pretreatment. Additionally, the functional enrichment exploration indicated that the high-risk agents were enriched in the ‘citrate cycle’ (TCA cycle) and ‘pyruvate metabolism’. Tsvetkov et al. observed that cells that were highly dependent on mitochondrial respiration were more sensitive to elesclomol treatment, leading to cells being more sensitive to copper-induced cell death, which indicated a tight correlation with the TCA cycle. Nevertheless, the potential roles of these genes in STS need to be investigated in future studies.

When conducting and validating the prognosis model, the consequences indicate that STS patients with significantly different CRG expression levels have different immune infiltration and active immunity patterns. The tumor microenvironment (TME) consists of core tumor cells, immune cells, and stromal cells, which communicate intercellularly via the release of chemokines and cytokines. An increasing body of proof indicates that patients who show a better immune-inflamed TME tend to have a better prognosis, especially in patients with advanced solid tumors who need a sustained immune response to expand their overall survival (OS) when undergoing immunotherapy. Therefore, it is necessary and crucial to further investigate the correlation and interaction between immune cells and TME. In our study, STS patients in the high-risk group indicated increased type 2 helper cell, but the majority of cases showed decreased cell types such as activated B cells, natural killer (NK) cells, CD^4/8+^ T cells, Th1/2 cells, eosinophil, dendritic cells, monocytes, neutrophils, and macrophages. Moreover, increased tumor-infiltrating immune cells were presented in the low-risk group, indicating that STS patients engaged with more immune cells have a better prognosis, which is consistent with previous research.

Immunotherapy prediction showed that the cuproptosis score could serve as a potential immunotherapy marker, which provides us a better understanding of cuproptosis and specific guidance for therapeutic and prognosis prediction of STS [36,37]. However, how CRGs have impacted the tumorigenesis of STS by recruiting the immune cells of TME has not been fully elucidated. A GO analysis was performed to investigate immunity and metabolic-related biological processes (BPs) that were closely correlated with the function of these CRGs between the high- and low-risk groups. Moreover, gene set enrichment analysis (GSEA) was performed on CRGs to identify their enrichment during immunity and metabolic-associated BPs, and the results of the heatmap showed that the immunity-associated BPs were active in the low-risk group versus the high-risk group. Therefore, analysis of immune checkpoints and tumor mutation burden analysis were carried out to further examine the efficiency of immune checkpoint therapy. The result showed that low-risk patients had increased levels of immune checkpoints such as PDCD1, CD274, CTLA4, CD86, PDCD1LG2, LAG3, HAVCR2, and CXCL9. Additionally, current studies indicate that a higher tumor immune dysfunction and exclusion (TIDE) score indicates patients in the high-risk group have a more resistant clinical response to checkpoint immunotherapy compared to the low-risk group. CYT stands for ‘cytolytic activity’, which was acquired through calculating the geometric mean of granzymes A (GZMA) and perforin (PRF1) mRNA expression levels in tissue. Additionally, the CYT score was correlated with cytotoxic T cell (CTL) markers and prognosis in pan-cancer TCGA datasets, reflecting host immune conditions [38]. In our study, the CYT score was lower in the high-risk group, which showed that a low CYT score could be a risk factor and worse prognosis. Therefore, the above results indirectly demonstrate that the patients in low-risk group might be more sensitive to immune checkpoint therapy, which it could serve an essential factor in regulating the clinical response to immunotherapy [39].

In the TCGA-Sarcoma cohort, mutation analysis was conducted by presenting that in the high-risk group compared to the low-risk group, showing the higher frequency mutated rate among especially for TP53, ATRX, and MUC16. Based on existing research, TP53 is the most prevalent mutant gene in cancer, and its mutation can regulate the capacity of p53 to improve physiological processes including apoptosis and iron-induced cell death. Moreover, recent whole-genome sequencing analyses have shown more frequent alterations in TP53, including structural alterations in TP53 intron 1, and the impacts of TP53 on STS progression have been underestimated. Thus, it is vital to further unveil the mechanism of TP53 in the suppression of bone and soft tissue sarcoma progression. As for ATRX and MUC16, the study of STS, or cuproptosis, is still needed to further investigate [39,40].

In this study, we performed a bioinformatic analysis to establish a prognostic model of STS patients and investigate the correlation between cuproptosis-related genes and immune cell infiltration via data from the TCGA database. Notwithstanding, there are several limitations in our study. First, though it was confirmed by qPCR to verify the results, it was lacking experiments for validating immune cells infiltration, which needed further study. Moreover, there are few supervised clustering methods designed especially for tumor genomics utilization, and therefore unsupervised clustering offers an option for us. Third, the immunotherapy data of STS patients were not available in our hospital, so there was no strong evidence to validate the bioinformatic results, and the predictive cuproptosis score for sarcoma is supposed to be further verified in the future. Last but not least, a greater reliability and validity of the cuproptosis score with an increased sample size and the number of validation sets will be conducted in order to reach more satisfactory consequences.

## 5. Conclusions

In summary, we first examined the expression level of CRGs in patients with qRT-PCR, verifying the results, and identified two cuproptosis-related subtypes in patients with sarcoma, with different prognoses, mutation characteristics, and immune infiltration. Additionally, a risk model termed the cuproptosis score was conducted to estimate the cuproptosis level of each group thoroughly, where high-risk scores showed poor prognosis and low immune-infiltration levels. We draw a graphical abstract to demonstrate the analysis workflows of this study. Notably, the cuproptosis combined immune infiltration-related risk score model indicates an immunological correlation on tumor progression and possible evidence in developing immunotherapy, which help clinicians choose the optimal therapeutic strategies for personalized therapy in clinical practice.

## Figures and Tables

**Figure 1 cells-11-04077-f001:**
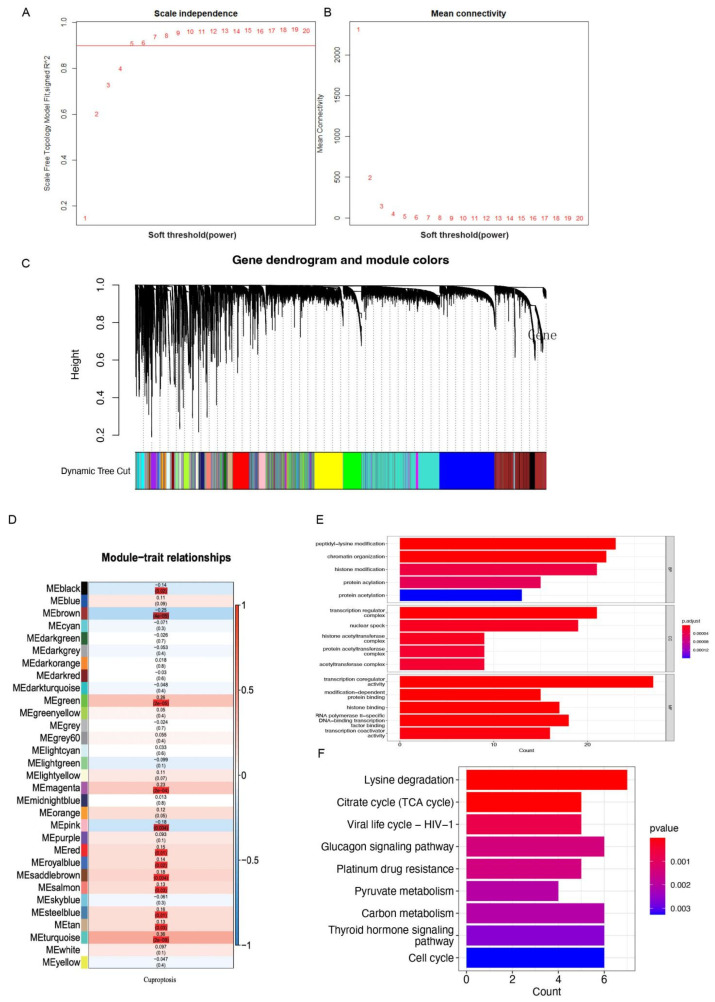
WGCNA revealed potential mechanisms and pathway enrichment analysis of CRGs in STS patients. (**A**,**B**) The distribution and tendency of scale-free topology model fit and mean connectivity accompanied by a soft threshold. (**C**) A dynamic tree cut was merged with a dynamic method to decipher the clustering of genes among different modules. (**D**) The mean correlations between module and cuproptosis. The colors of the cells represent associated intensity, and the numbers in parentheses represent the *p* value of the correlation test. (**E**) the enriched item in the gene ontology analysis; (**F**) the enriched item in the Kyoto Encyclopedia of Genes and Genomes analysis. The size of the circles represents the number of enriched genes. BP: biological process; CC: cellular component; MF: molecular function; and CRG: cuproptosis-related gene.

**Figure 2 cells-11-04077-f002:**
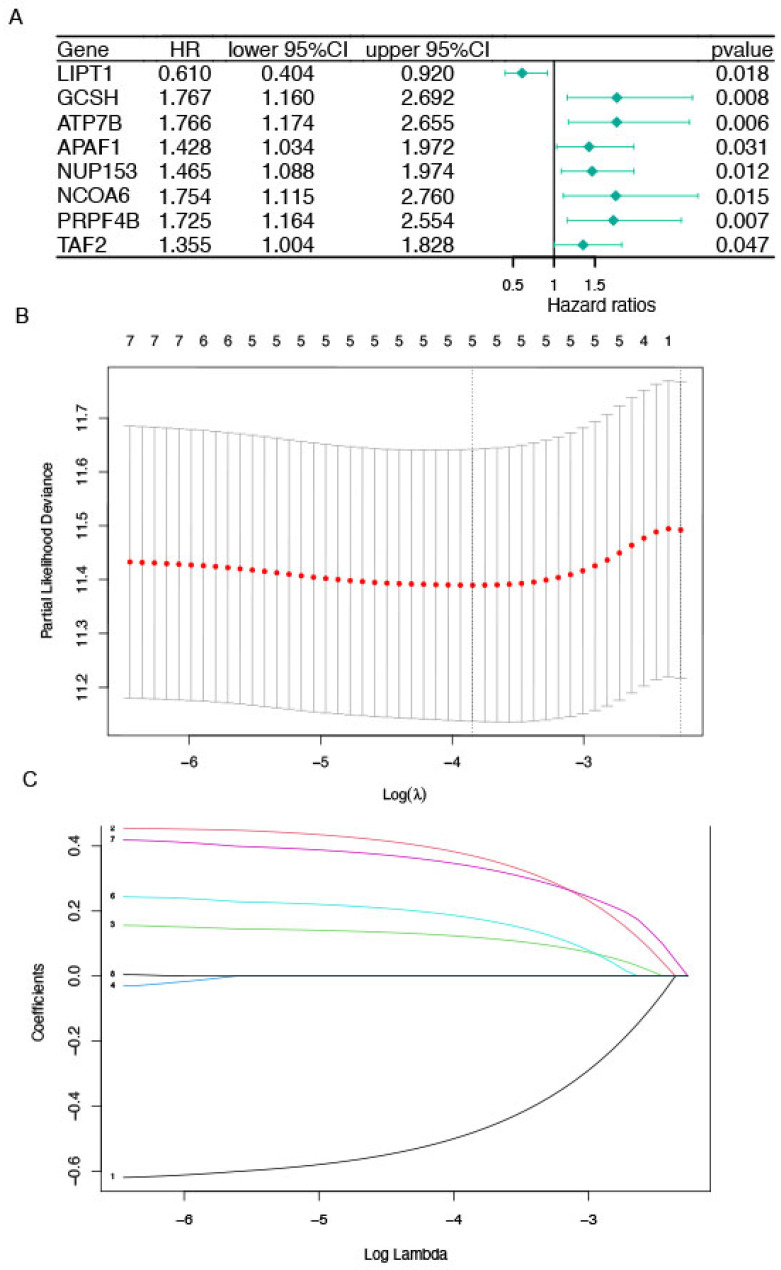
Construction and validation of prognosis-associated genes model of STS using LASSO regression. (**A**) The result of cox regression analysis of prognostic genes for patients with sarcoma. (**B**,**C**) the results of LASSO regression model.

**Figure 3 cells-11-04077-f003:**
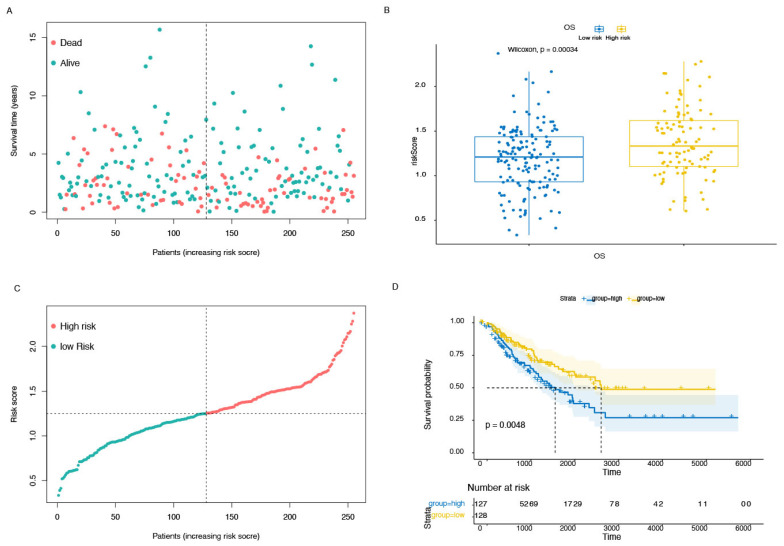
The association survival difference between high- and low-risk groups. (**A**,**B**) The risk curves and the distribution of patients are ordered according to the risk scores, from low to high. The dots in the lower part indicate the distribution of cases. (**C**) The correlation between overall survival and risk score. (**D**) Survival differences between the high- and low-risk groups in patients with sarcoma from the database. The table below the survival curves shows the number of patients alive in each year.

**Figure 4 cells-11-04077-f004:**
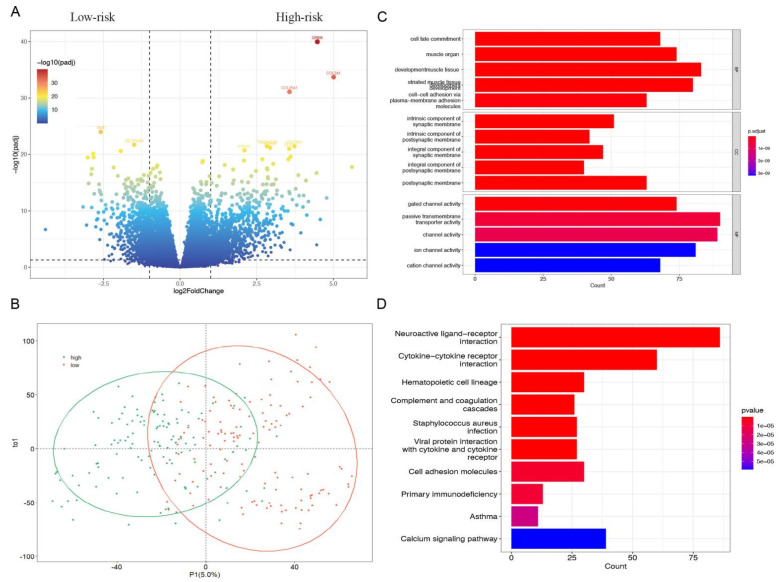
Exploring of DEGs of high- and low-risk groups in patients with sarcoma. (**A**) The result of a volcano plot. The dots on the right site indicated the high-risk genes, the dot on the left site is the low-risk genes, and the in-between dots indicated the other genes without significant difference. (**B**) The result of PLS-DA indicated the two groups of people could be clearly distinguished in the model. (**C**,**D**) The GO functional enrichment analysis of differential genes includes three domains: molecular function (MF), biological process (BP), and cell composition (CC). KEGG pathway analysis of differentially expressed genes.

**Figure 5 cells-11-04077-f005:**
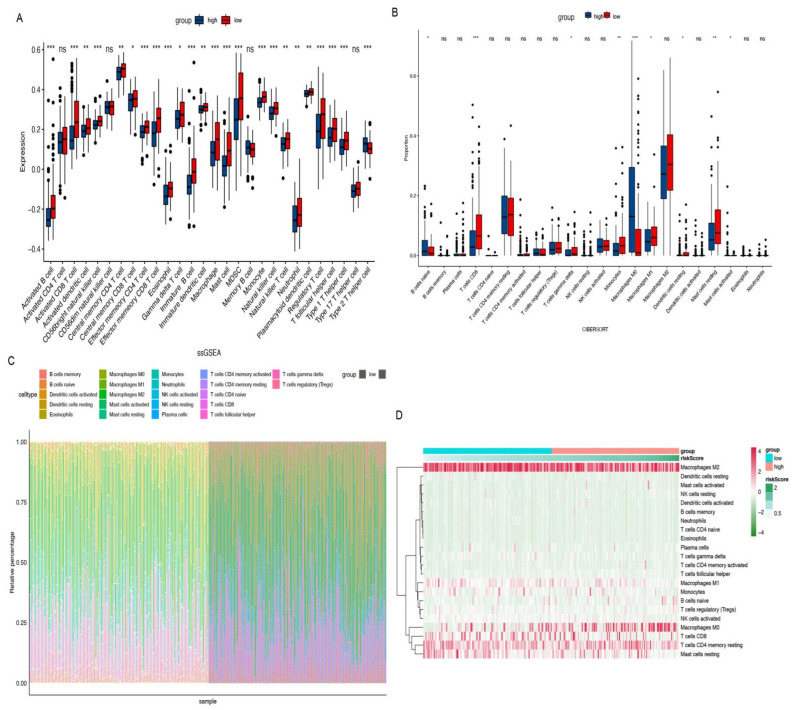
Immune cell populations and TME landscape. (**A**,**B**) The result of ssGSEA and CIBESORT to demonstrate the difference of immune cell populations between the high- and low-risk groups. (**C**,**D**) The association and proportions of risk and immune-infiltrations. * *p* < 0.05. ** *p* < 0.01. *** *p* < 0.001. ns: not significant.

**Figure 6 cells-11-04077-f006:**
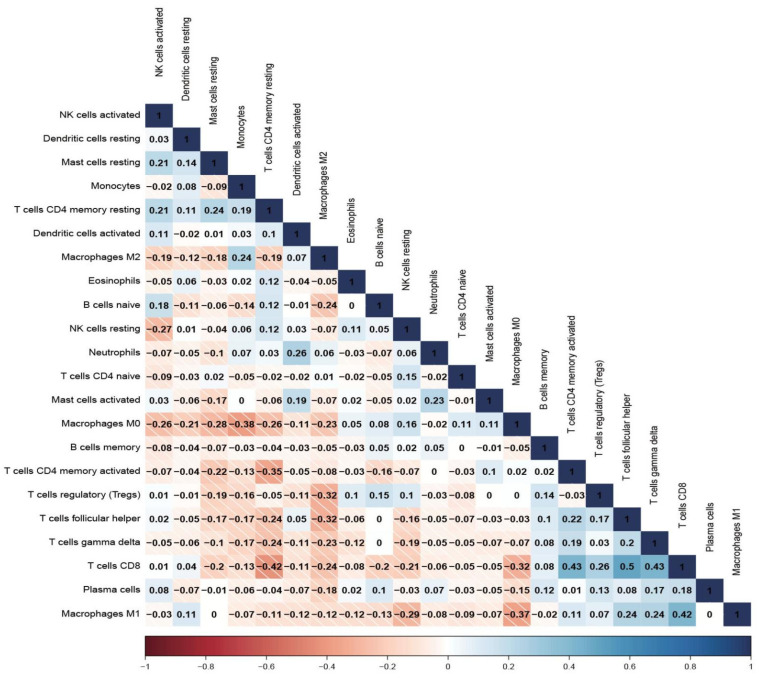
The correlation between different immune cells.

**Figure 7 cells-11-04077-f007:**
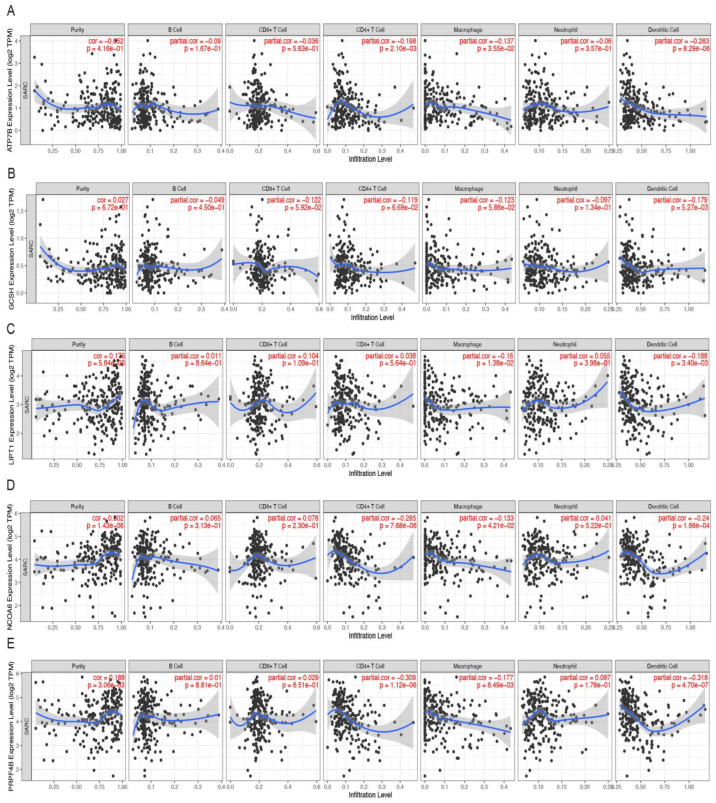
Relationships between expression of five hub genes and tumor immune infiltrations. (**A**) LIPT1, (**B**) GCSH, (**C**) ATP7B, (**D**) NCOA6, (**E**) PRPF4B.

**Figure 8 cells-11-04077-f008:**
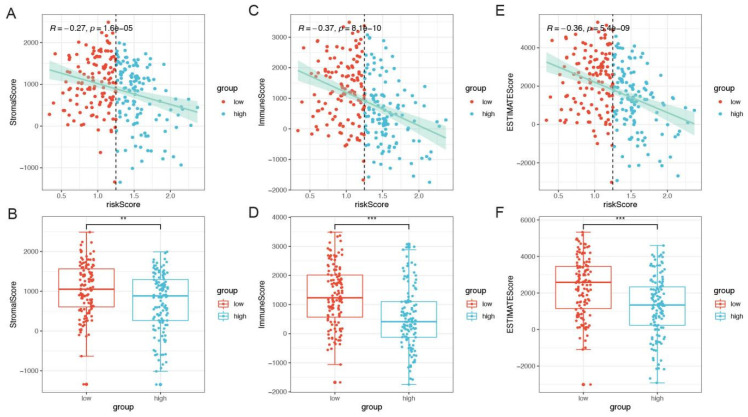
The association of CRGs with immune scores. (**A**–**F**) Immune scores, stromal scores, and ESTIMATE scores of five CRGs signatures between the high- and low-risk groups. Immune and stromal scores were calculated by analyzing specific gene expression signatures of immune and stromal cells to predict non-tumor cell infiltration. ** *p* < 0.01. *** *p* < 0.001.

**Figure 9 cells-11-04077-f009:**
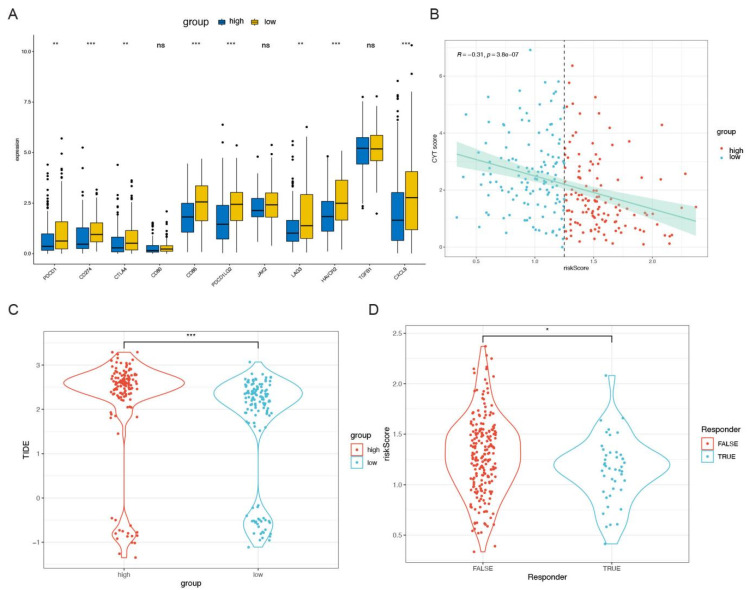
The association of CRGs with immune checkpoints predicting the response to chemotherapy. (**A**) The differences in expression levels of immune checkpoints between the in patients with sarcoma. (**B**–**D**) The results of CYT, TIDE1, and TIDE2. CYT: cytolytic activity, TIDE: tumor immune dysfunction and exclusion. * *p* < 0.05. ** *p* < 0.01. *** *p* < 0.001. ns: not significant.

**Figure 10 cells-11-04077-f010:**
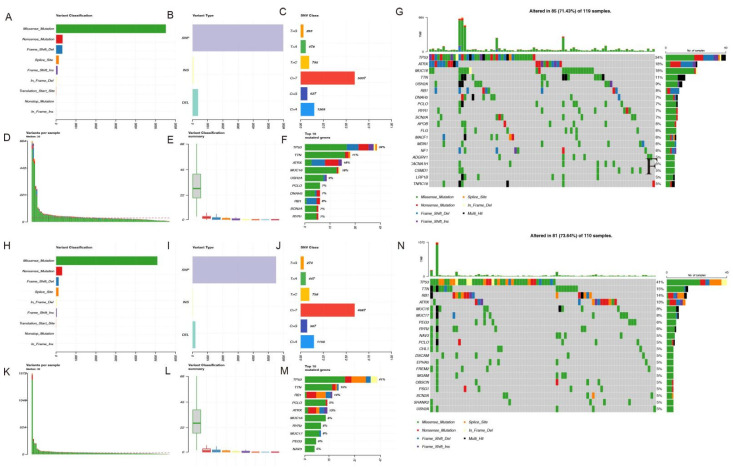
Mutation landscape. (**A**–**H**) Variant classification. X axis indicated variant numbers. Y axis showed different variant classifications. (**B**–**I**) Variant type. X axis indicated variant numbers. Y axis represented different variant types. (**C**–**J**) SNVs type. X axis indicated the ratio. Y axis represented the type of nucleotide substitution. (**D**–**E**, **K**–**L**) The variants per sample and the summary of variant classification. Each sample contains a statistical plot of the number of mutations and a box plot of the classification of the various mutations in the sample (**F**–**M**) Top10 mutated genes. X axis indicated variant numbers. Y axis represented different genes. The genes were ordered by their mutation frequency.) (**G**–**N**) The upper bar shows the total gene mutation amount and corresponding mutation types. The right bar shows the mutation frequency of the top 20 mutated genes.

**Figure 11 cells-11-04077-f011:**
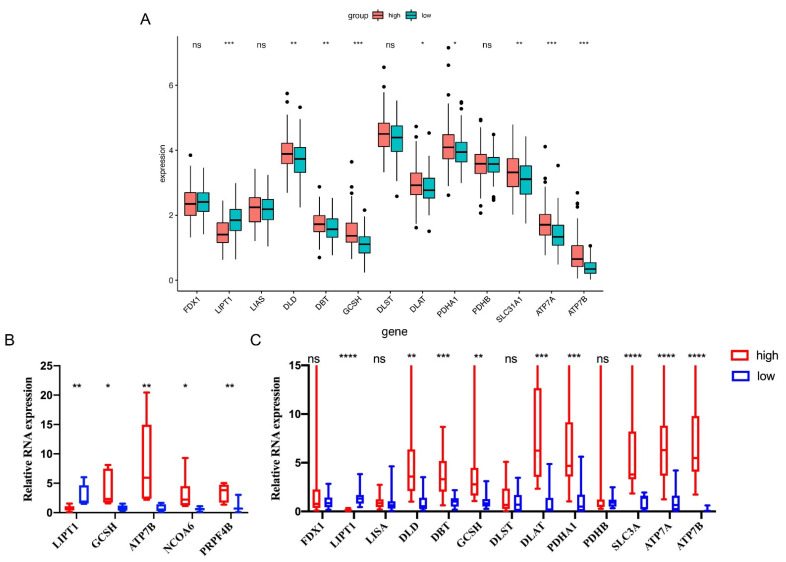
Relative expression level of selected CRGs in sarcoma tissues. The expression level of CRGs in high- and low-risk groups of patients with sarcoma by informatic analysis. (**A**) Validation of the expression of LIPT1, GSCH, ATP7B, NCOA6, and PRPF4B in sarcoma tissue between two groups (**B**) The expression level of LIPT1, DLD, DBT, DLAT, PDHA1, PDHB, SCL31A1, ATP7A, and ATP7B between the high- and low-risk groups (**C**). GAPDH was used as the internal reference gene for qRT-PCR relative expression. Error bars indicate the standard deviation or the standard error of the data. * *p* < 0.05. ** *p* < 0.01. *** *p* < 0.001. **** *p* < 0.0001. ns: not significant.

## Data Availability

All data generated or analyzed during the current study are available from the corresponding author on reasonable request. Publicly available datasets were analyzed in this study. This data can be found here: TCGA database (https://portal.gdc.cancer.gov) Accessed on 5 May 2022. Single sample gene-set enrichment analysis (ssGSEA) was conducted based on the expression level of 28 immunity-associated signatures (http://cis.hku.hk/TISIDB/download.php) accessed on 5 May 2022. The PCR data that support the findings of this study are available from the corresponding author, Anquan Shang and Guodong Li, upon reasonable request.

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
