# Peer review of "A Combined Risk Score Model to Assess Prognostic Value in Patients with Soft Tissue Sarcomas"

_cells, 2022, doi:10.3390/cells11244077_

Round 1

Reviewer 1 Report (Previous Reviewer 2)

Title must be changed: No data is provided on the role of cuproptosis in soft tissue sarcoma DEVELOPMENT.

Authors do not suggest any rationale on why cuproptosis associates to different immune infiltrate.

Authors may wish to add evaluation of cuproptosis risk score along with primary diagnosis of TCGA sarcoma cohort.

Legends to figures are less descriptive than in the first submission, while the request was to have them more descriptive to enable readers to understand what they were looking at. Just to make an example: panel A of figure 7 what represents? Total gene mutation ….. in which group? Besides, figure 7 is illegible. Try to split it in 2 figures, possibly deciding to put part of the information in supplementary files.

Line 89. “autography”, could authors expand on how “autography” can induce cell death?

Line 91. “the expression of Cu could”. Cu is a metal, the term expression is not adequate.

Lines 102-105. “Given the tight correlation between copper homeostasis and tumorigenesis, and therefore, it is vital to recognize CRGs and to ….”. What’s the use of “and therefore” here?

Line 108. “while also attempting to assess alternative molecular mechanisms,” the sentence is not clear. Do the authors want to investigate: i) the mechanisms through which cuproptosis impacts on prognosis and immune infiltration of STS; ii) other mechanisms through which cuproptosis impacts STS; iii) other mechanisms, other than cuproptosis, that affect prognosis and immune infiltrate of STS. All the sentence, till the full stop, needs to be rewritten.

Lines 124-126. Should be removed from here and integrated in the section describing the evaluation of immune infiltrate and tumor microenvironment.

Line 328. The authors refer to “Differentially Expressed Genes in High- and Low-Risk Group”, however it is not clear how they obtain it, with which analysis and package(s).

Line 331-332. “|log Fold Change| > 0.5)”, which is the base?.

Line 371-372. “Additionally, the results of risk scores wielded negative correlations with vast majority of tumor infiltration with the utilization of CIBERSORT algorithm and deleting data with no statistically significant difference at the same time (Figure 4C).” Please rephrase.

Lines 391-396. “According to the consequences, plasma cell, T cell gamma delta, macrophages, mast cells resting, and T cells CD8, 4 memory resting, etc. was notable, which reveal that lower infiltration enrichment in the high-risk subtype while the mast cells resting presented higher level in the low-risk subtype and macrophages M0 presented higher levels in the high-risk subtype (Figure 4E).” Please rephrase. What is the take home message, any statistically significant difference between high and low risk groups?

Lines 408-409.” Based on the knowledge that TME consisting of multiple immune cells have significant impacts on the prognosis of STS, the correlation was investigated.” Authors should state something like: the correlation between immune cell infiltrate and the expression of the 5 hub CRGs was investigated. If I understood correctly.

Line 426. “RBPs”. Please spell out or correct.

Lines 470-472. “The results showed that the DEGs level of LIPT1, DLD, DBT, DLAT, PDHA1, PDHB, SCL31A1, ATP7A and ATP7B between the high- and low-risk group by bioinformatic analysis. (P < 0.05, Figure 8A). Please rephrase.

Line 437. Appreciate that the authors added the detailed information on what “CYT” is in the discussion. However, it would be much better to have it at the point in which the data are presented, at least the description of what CYT is, and leave its role in prognosis and CTL infiltration in the discussion.

Line 441. “TRUE and FALSE group”, please expand. What represents true and false and what is the information provided. Responders to what? The reader needs to understand the data they are presented here, and not wait until the discussion to learn that response is to ICI.

Lines 444-445. “immune risk score”. I am confused, I understood that the risk score was made up on cuproptosis genes. If I understood correctly, please rephrase this point.

Lines 472-474. Authors must state HERE on what series they performed RTqPCR validation (guess the 33 patients refered to in lines 204-211).

Figure 2. Authors have changed the labeling in Figure 2F. The legend for OS changed from 0 and 1 to low and high risk, following the reviewers question. But that was a question. Is it correct, or is the data referring to dead/alive? Please check carefully. Just as a sidenote, text in the figure (number in the axis, text in the table) is messy, shown as symbols in my PC. Same problem for y axis labels of Figure 1 E, F.

E and F referencing is now absent in legend to figure 2.

Figure 4C. Since legend is not clear, I do not understand what is represented in Figure 4C, nor I am able to tell it from the text. As is, I read it as a correlation analysis among the different immune cells in the samples analyzed, but I do not perceive the role of the risk factor. Plus, the materials and methods section should report wow the correlation was calculated.

Figure 4D is still difficult to read. If nothing can be done to further improve it, then leave it as it is. Legend for grouping appears to be cut.

Supplementary Table 2 lacks tumor type, i.e. primary diagnosis as reported in Supplementary Table 4.

Author Response

Reviewer 2 Report (Previous Reviewer 3)

Dear Authors,

the immunofluorescence images included in response letter do not address the issue of external validation. 

As for the other issues, I consider the the changes satisfactory.

Author Response

This manuscript is a resubmission of an earlier submission. The following is a list of the peer review reports and author responses from that submission.

Round 1

Reviewer 1 Report

The content of this paper is comprehensive and the field may be useful for the management of soft tissue sarcoma patients. I recommend it for publication. In the complex manuscript is well written, however I I believe that it would be scientifically more acceptable to extend the validation series to a number greater than 33 STS patiens

Author Response

Dear Reviewer,

We thank you for the thoughtful and helpful comments on our original manuscript. We have now revised the manuscript, taking into account the Reviewers' constructive comments and suggestions. Specifically, we have attached the supplementary data of the levels of lesions. We have also included the contents in the revised manuscript to further discuss the current results.

Anquan Shang

Q1: The content of this paper is comprehensive and the field may be useful for the management of soft tissue sarcoma patients. I recommend it for publication. In the complex manuscript is well written, however I I believe that it would be scientifically more acceptable to extend the validation series to a number greater than 33 STS patiens

Response: We thank the reviewer for the affirmation and will make revisions according to comments. Further research in this direction will be carried out in the future.

Reviewer 2 Report

Authors evaluated the prognostic role of a cuproptosis gene signature in soft tissue sarcomas and its relation to immune cell infiltration. The study is correlative and lacks adequate validation in an independent cohort. Validation set is too small, and in this set authors just show a different expression of CRG between high and low risk patients, but not the prognostic significance of the CRG-based signature. English language is poor, which makes extremely difficult to follow the paper. Would the authors suggest a mechanism by which CRG genes affect survival and immune infiltration in soft tissue sarcomas?

Specific comments

All the text needs an extensive language revision in order to allow readers to clearly understand the contents of the paper, and the methodology used.

Title: “The Role of Cuproptosis-Related Genes in the Development, Tumor-Associated Immune Cell and Prognosis of Soft Tissue Sarcoma.” No data is provided on the role of cuproptosis in soft tissue sarcoma development, please rephrase.

All figure legends must be more detailed, enough to allow the reader to understand what he is looking at.

Name of files and title of tables herein contained do not match: Supplementary file labeled “Supplementary Table 3” contains table entitled “Supplementary Table 2”; Supplementary file labeled “Supplementary Table 2” contains table entitled “Supplementary Table 4 Differential Expressed Genes”; Supplementary file labeled “Supplementary Table 4” contains table entitled “Supplementary Table 5. Patient Characteristics” (with 2 sheets, which one is correct?); Supplementary file labeled “Supplementary Table 5” contains table entitled “Supplementary Table 3 The primer sequences of genes”

The first table to be cited in the text is Supplementary Table 3.

All supplementary tables must have a short legend and explicative title to allow readers understand what it is presented. E.g. what is reported in table entitled Supplementary Table 2? What are “Level, High, Low, P value” in Supplementary Table entitled “Supplementary Table 1 Charcteristic”.

Materials and methods.

Add package and software versions used (e.g. limma, clusterprofiler). What tools were used to perform GSVA and WGCNA?

Report how the mutational landscape of sarcoma samples analyzed was obtained.

Authors must state what parameter was used for evaluating prognosis. From figures it appears to be overall survival, but it must be clearly stated in the text both in materials and methods and in the results sections.

Page 4. Line 145, what signatures were used for evaluating immune infiltrate with ssGSEA?

Results

Page 5, lines 195-196. Authors state “The module with red color showed a slightly positive relationship with survival status (Figure 1C)”. How could readers recognize this information from the referenced panel?

Page 9, lines 284-285. Data presented in figure 5 show the association of immune infiltrate to the levels of each of the 5 CRGs included in the prognostic signature and not to the risk group/level.

Page 12, lines 326-327 and Figure 6H, what CYT score stands for?

Paragraph 3.5. Hub CRGs Immune Infiltration Analysis. Not clear the rational of dissecting the contribution of each CRG, what adds vs. the CRG score?

3.8. qRT-PCR Validations in Sarcoma Patients’ Tissues. It is not clear from which samples the data were derived, are those of the 33 pts cohort? Authors should clearly state.

Figure 1 entitles “WGCNA was performed to identify the potential mechanisms associated with the prognostic signature of CRGs in patients with sarcoma”. However, to me the link with prognosis here is not evident. Authors must render it clear to the general reader. Besides, authors should highlight significant correlations in Figure 1D.

Legend to Figure 1E,F. Authors should state which letter refers to which analysis.

Figure 2. Legend must be more descriptive. In the figure, authors must use consistent labeling (e.g. does the groups labeled 0 and 1 in Figure 2F correspond to low and high risk?).

Figure 3. What is the direction of the contrast: high-risk vs low risk or the opposite?

Figure 4C appears to report correlations among immune cell populations, and not the association between risk and immune cell infiltration as suggested by the legend and the results.

Figure 4D data presented in this format are illegible.

Figure 6J. x-axis and legend, “responder” to what?

Legend to Figure 8. “Error bars indicate the standard deviation or the standard error of the data.” Authors must state where the bars refer to standard error and where to standard deviation.

Author Response

Dear Reviewer,

We thank you for the thoughtful and helpful comments on our original manuscript. For the problems mentions in involved Figure, we apologized for not arraying and explaining well. Therefore, we thanked for the reviewer’s suggestion and revised the figures, and then re-uploaded them.

Anquan Shang

Q1: All figure legends must be more detailed, enough to allow the reader to understand what he is looking at. 

Response: We thank the Reviewer for the good comments and advice. We have modified the figure legends, making it more concise, accurate and understandable.

Q2: Name of files and title of tables herein contained do not match: Supplementary file labeled “Supplementary Table 3” contains table entitled “Supplementary Table 2”; Supplementary file labeled “Supplementary Table 2” contains table entitled “Supplementary Table 4 Differential Expressed Genes”; Supplementary file labeled “Supplementary Table 4” contains table entitled “Supplementary Table 5. Patient Characteristics” (with 2 sheets, which one is correct?); Supplementary file labeled “Supplementary Table 5” contains table entitled “Supplementary Table 3 The primer sequences of genes” ; The first table to be cited in the text is Supplementary Table 3.

Response: We are very sorry for this error; We have checked all the file names and re-uploaded.

Q3: All supplementary tables must have a short legend and explicative title to allow readers understand what it is presented. E.g. what is reported in table entitled Supplementary Table 2? What are “Level, High, Low, P value” in Supplementary Table entitled “Supplementary Table 1 Charcteristic”.

Response: We have revised the supplementary tables according to the Reviewer’s suggestions. We changed ‘Charcteristic’ to ‘Patient Characteristics’. ‘High, low’ means high risk group and low risk group. P value means the results according to statistical analysis.

Materials and methods

Q4: Add package and software versions used (e.g. limma, clusterprofiler). What tools were used to perform GSVA and WGCNA?

Response: clusterprofiler (version 4.4.4), limma (version 3.52.3), GSVA package (version 1.22.4), WGCNA package (version 1.70). We use R software to perform GSVA and WGCNA.

Q5: Report how the mutational landscape of sarcoma samples analyzed was obtained.

Response: RNA-sequencing expression profiles and corresponding clinical information for SARC were downloaded from the TCGA dataset(https://portal.gdc.com). TMB is derived from the article The Immune Landscape of Cancer published by Vesteinn Thorsson et al. in 2018; MSI is derived from the Landscape of Microsatellite Instability Across 39 Cancer Types article published by Russell Bonneville et al. in 2017. Data analysis was performed by maftools package (Version2.12.0).

Reference:

Thorsson V, Gibbs DL, Brown SD, et al. The Immune Landscape of Cancer [published correction appears in Immunity. 2019 Aug 20;51(2):411-412]. Immunity. 2018;48(4):812-830.e14. Russell, Bonneville, Melanie, et al. Landscape of Microsatellite Instability Across 39 Cancer Types.[J]. JCO precision oncology, 2017. Frost FG, Cherukuri PF, Milanovich S, Boerkoel CF. Pan-cancer RNA-seq data stratifies tumours by some hallmarks of cancer. J Cell Mol Med. 2020;24(1):418-430. Izzi V, Davis MN, Naba A. Pan-Cancer Analysis of the Genomic Alterations and Mutations of the Matrisome. Cancers (Basel). 2020;12(8):2046. Published 2020 Jul 24. Zhang Q, Huang R, Hu H, et al. Integrative Analysis of Hypoxia-Associated Signature in Pan-Cancer [published online ahead of print, 2020 Aug 14. iScience. 2020;23(9):101460.

Q6: Authors must state what parameter was used for evaluating prognosis. From figures it appears to be overall survival, but it must be clearly stated in the text both in materials and methods and in the results sections.

Response: We apologize for not correctly explaining for the method of evaluating the prognosis according to the data from TCGA. Patients were divided into high risk and low risk groups based on LASSO regression gene expression. Survival analysis was performed according to the patient's end point and survival time.

Q7: Page 4. Line 145, what signatures were used for evaluating immune infiltrate with ssGSEA?

Response: Single sample gene-set enrichment analysis (ssGSEA) was conducted based on the expression level of 28 immunity-associated signatures ((http://cis.hku.hk/TISIDB/download.php) (See the supplementary file in detail cellMarkers).

Result:

Q8: Page 5, lines 195-196. Authors state “The module with red color showed a slightly positive relationship with survival status (Figure 1C)”. How could readers recognize this information from the referenced panel?

Response: We apologize for not correctly explaining for the figure. ‘The module with red color showed a slightly positive relationship with survival status’ This sentence should be explained for Figure 1D. As for Figure 1C, we have revised in the paper with ‘The gene clustering results were cut to obtain different gene modules, and each gene module was assigned a color value. Finally, the results were added to the clustering tree, and the color assigned previously was used for differentiation’

Q9: Page 9, lines 284-285. Data presented in figure 5 show the association of immune infiltrate to the levels of each of the 5 CRGs included in the prognostic signature and not to the risk group/level.

Response: We apologize for not correctly explaining for the association between the hub genes and immune infiltration level. We have revised the paper as ‘The proportions of immune Infiltration from 22 cell types in the two risk subtypes were presented in a bar plot (Figure 4D). In order to investigate the association of immune cells between the signature genes levels, the levels of tumor immune infiltration were conducted, which was presented in Figure 5.’

Q10: lines 326-327 and Figure 6H, what CYT score stands for?

Response: We apologize for not correctly explaining for the definition of CYT score. CYT stands for ‘cytolytic activity’, which is strongly correlated with host immune conditions and reflected prognosis according to an article published in Cell ‘Molecular and genetic properties of tumors associated with local immune cytolytic activity’, which was also added into the references. (doi:10.1016/j.cell.2014.12.033.) Additionally, an published article ‘Cytolytic Activity (CYT) Score Is a Prognostic Biomarker Reflecting Host Immune Status in Hepatocellular Carcinoma (HCC)’  could be serve as a reference.(doi: 10.21873/anticanres.13030)

Q11: 3.8. qRT-PCR Validations in Sarcoma Patients’ Tissues. It is not clear from which samples the data were derived, are those of the 33 pts cohort? Authors should clearly state.

Response: We feel very apologize for not explaining this part clearly. Now we have supplemented it in the revised version. ‘A total of 33 STS patients’ who underwent radical resection at the Shanghai Tongji Hospital and Shanghai Tenth People’s Hospital of Tongji University. None of the patients underwent preoperative radiotherapy or chemotherapy. According to TNM classification, we divided STS patients enrolled into high-risk group and low-risk group, including 20 high-risk and 13 low-risk patients. The clinicopathological details are presented in Additional File: Supplementary Table 2. Ethical approval for the study was granted by the Clinical Research Ethics Committee in our hospital and all participants included in our study provided written informed consents.’

Q11: Figure 6J. x-axis and legend, “responder” to what?

Response: Based on tumor pre-treatment expression profiles, this TIDE module can estimate multiple published transcriptomic biomarkers to predict patient response.

Reviewer 3 Report

This study aimed to evaluate the role of cuproptosis-related genes (CRG) in soft tissue sarcomas. The study was essentially based on public data from the TCGA-SARC series. Authors developed a prognostic model and defined a Cuproptotic Risk Score based on the expression of 5 CRG.  Authors also evaluated tumour immune infiltration, stromal component and immune cell function by different bioinformatic tools.  The Cuproptotic Risk Score was inversely correlated with immune infiltration. In another smaller series of soft tissue sarcomas, CRGs expression was evaluated by qRT-PCR and correlated with risk.

A major problem of this study is the validation of the performance of the prediction model. In the present study the model was validated only in the same series in which the model was generated. In other words, an external validation is lacking. The qRT-PCR data in the Shangai series is not an appropriate validation of the model but is only be representative of the expression pattern for the selected CRG genes.

Another problem is that there isn’t any cross validation of immune infiltration results, therefore all the discussion about correlations with immunotherapy is speculative.  

Some references are mis-citations or are not relevant to this study: ref 19 is a paper about origin and transmission of COVID-19; ref 37 is a study about skin squamous cell carcinomas and not about soft tissue sarcoma; ref 46 (cited at page 15) is not the paper by Kakuda et al. but another publication.

The legend to FIG 6 is incomplete because description of H, I and J are missing.

In FIG 7 the text font sizes are too small and it is very difficult to read the text. In the legend to FIG 7 “E-G” are described but there are not E-G panels.

In section 3.8 authors report the evaluation of expression by “bioinformatic analysis” This is too generic. Authors should detail how expression was evaluated. In FIG 8A the units of expression should be specified (TPM? RPKM?). Moreover, it is not clear which samples were evaluated in experiment displayed in FIG 8B.

Finally, the text needs English language improvement because there are many errors

Author Response

Dear Reviewer,

We thank you for the thoughtful and helpful comments on our original manuscript. We have included the contents in the revised manuscript to further discuss the current results.

Anquan Shang

Q1: A major problem of this study is the validation of the performance of the prediction model. In the present study the model was validated only in the same series in which the model was generated. In other words, an external validation is lacking. The qRT-PCR data in the Shangai series is not an appropriate validation of the model but is only be representative of the expression pattern for the selected CRG genes. Another problem is that there isn’t any cross validation of immune infiltration results, therefore all the discussion about correlations with immunotherapy is speculative.  

Response: Thanks for the good suggestions. The lack of external validation set in this study is a shortcoming for us. We are also working to address this issue. Regarding the results of immune infiltration, it is indeed just an existential phenomenon and, like your interpretation, is speculative to us and we have re-edited the language to describe it.

Q2: Some references are mis-citations or are not relevant to this study: ref 19 is a paper about origin and transmission of COVID-19; ref 37 is a study about skin squamous cell carcinomas and not about soft tissue sarcoma; ref 46 (cited at page 15) is not the paper by Kakuda et al. but another publication.

Response: Thanks for the good suggestions. We have checked and deleted references that are not relevant to the main content.

Q3: The legend to FIG 6 is incomplete because description of H, I and J are missing. In FIG 7 the text font sizes are too small and it is very difficult to read the text. In the legend to FIG 7 “E-G” are described but there are not E-G panels.

Response: We apologize for not correctly clarifying the figures. Due to uploading the wrong version of the figures, a misunderstanding was caused and now they had been revised and re-uploaded. For the difficult to read the content of the figure, we have re-uploaded the high-resolution reversion of the figure.

Q4: In section 3.8 authors report the evaluation of expression by “bioinformatic analysis” This is too generic. Authors should detail how expression was evaluated. In FIG 8A the units of expression should be specified (TPM? RPKM?). Moreover, it is not clear which samples were evaluated in experiment displayed in FIG 8B.

Response: We apologize for not correctly clarifying the details of the figure. The discrepancy expression of each gene was based on FPKM, and more details was uploaded as supplementary file genes expression. In Fig 8B, we performed qRT-PCR to examined and validate the result of LASSO regression. The details of the sample were supplemented in the revised version as ‘A total of 33 STS patients’ who underwent radical resection at the Shanghai Tongji Hospital and Shanghai Tenth People’s Hospital of Tongji University. None of the patients underwent preoperative radiotherapy or chemotherapy. According to TNM classification, we divided STS patients enrolled into high-risk group and low-risk group, including 20 high-risk and 13 low-risk patients. The clinicopathological details are presented in Additional File: Supplementary Table 2. Ethical approval for the study was granted by the Clinical Research Ethics Committee in our hospital and all participants included in our study provided written informed consents.

Q5: Finally, the text needs English language improvement because there are many errors

Response: We apologize for making the languages error. Errors were carefully checked and corrected, and native English speakers were invited for a second check and revision.

Round 2

Reviewer 3 Report

The major problem raised (the lack of model external validation) is still present. Authors response is rather elusive. 

There are still problems in Figures. In FIGURE 1E, 1F and FIG 2A several text characters are not readable.

There is a complete confusion in supplementary tables: file names do not correspond to file content.

English language still needs improvement.
